# Hepatocyte Growth Factor Enhances Antineoplastic Effect of 5-Fluorouracil by Increasing UPP1 Expression in HepG2 Cells

**DOI:** 10.3390/ijms23169108

**Published:** 2022-08-14

**Authors:** Manabu Okumura, Tomomi Iwakiri, Naoki Yoshikawa, Takao Nagatomo, Takanori Ayabe, Isao Tsuneyoshi, Ryuji Ikeda

**Affiliations:** 1Department of Pharmacy, Faculty of Medicine, University of Miyazaki, Miyazaki 889-1692, Japan; 2Department of Patient Safety Management, Miyazaki University Hospital, Miyazaki 889-1692, Japan; 3Division of Thoracic and Breast Surgery, Department of Surgery, Faculty of Medicine, University of Miyazaki, Miyazaki 889-1692, Japan; 4Department of Anesthesiology and Intensive Care, Faculty of Medicine, University of Miyazaki, Miyazaki 889-1692, Japan

**Keywords:** hepatocyte growth factor, HGF, 5-FU, UPP1, c-Met, erlotinib

## Abstract

Aberrant activation of hepatocyte growth factor (HGF) and its receptor c-Met axis promotes tumor growth. Therefore, many clinical trials have been conducted. A phase 3 trial investigating a monoclonal antibody targeting HGF in combination with fluoropyrimidine-based chemotherapy had to be terminated prematurely; however, the reason behind the failure remains poorly defined. In this study, we investigated the influence of HGF on the antineoplastic effects of 5-fluorouracil (5-FU), a fluoropyrimidine, in HepG2 cells. HGF suppressed the proliferative activity of cells concomitantly treated with 5-FU more robustly as compared to that of cells treated with 5-FU alone, and markedly increased the expression of uridine phosphorylase 1 (UPP1). Intracellular concentration of 5-fluorouridine, an initial anabolite of 5-FU catalyzed by UPP1, was increased by HGF. Interestingly, erlotinib enhanced HGF-induced increase in *UPP1* mRNA; in contrast, gefitinib suppressed it. Furthermore, erlotinib suppressed HGF-increased phosphorylation of the epidermal growth factor receptor at the Tyr1173 site involved in downregulation of extracellular signal-regulated kinase (Erk) activation, and enhanced the HGF-increased phosphorylation of Erk. Collectively, these findings suggest that inhibition of the HGF/c-Met axis diminishes the effects of fluoropyrimidine through downregulation of UPP1 expression. Therefore, extreme caution must be exercised in terms of patient safety while offering chemotherapy comprising fluoropyrimidine concomitantly with inhibitors of the HGF/c-Met axis.

## 1. Introduction

Hepatocyte growth factor (HGF) mediates multiple important cellular functions involved in tumor growth and development such as differentiation, proliferation, and migration via its receptor c-Met. Therefore, the aberrant activation of the HGF/c-Met axis is often associated with invasion, metastasis, disease stage, and shorter survival in multiple human cancer types. c-Met overexpression has been reported in many human cancers, such as non-small cell lung cancer (NSCLC) [1], gastric cancer [2], and hepatocellular carcinoma (HCC) [3]. Furthermore, c-Met gene amplification resulting in protein overexpression and constitutive activation of c-Met has also been described in the aforementioned cancer types [4,5,6]. Therefore, agents targeting HGF and c-Met are considered possible therapeutic candidates, and have been vigorously investigated [7,8,9,10]. Furthermore, a bispecific antibody targeting c-Met and the epidermal growth factor receptor (EGFR) has been recently studied [11]. However, concomitant administration of rilotumumab, a fully human monoclonal antibody targeting HGF, with fluoropyrimidine-based chemotherapy has been recently reported to be ineffective in improving clinical outcomes in patients with c-Met -positive carcinoma [12,13]. Among these trials, RILOMET-1, a phase 3 trial, had to be terminated prematurely, owing to the increasing number of deaths due to disease progression in the group of patients treated with rilotumumab concomitantly [12].

One reason for the disappointing results of the aforementioned trial is likely to be due to pharmacokinetic interactions between rilotumumab and fluoropyrimidine-based chemotherapy, especially with respect to metabolism. Additional and/or synergistic adverse events induced by the concomitant administration of rilotumumab with fluoropyrimidine-based chemotherapy have also been implicated. In fact, HGF, a target of rilotumumab, has been demonstrated to downregulate the expression of a critical enzyme, cytochrome P450 (CYP), which is involved in the metabolism of several drugs including antineoplastic agents [14], resulting in the occurrence of severe adverse events. Furthermore, we have previously reported that HGF suppresses the antineoplastic effect of irinotecan (CPT-11) by changing the expression of metabolic enzymes responsible for the conversion of CPT-11 to an active metabolite SN-38, which is subsequently converted to an inactive metabolite SN-38G [15]. Therefore, these findings, coupled with the fact that rilotumumab is a monoclonal antibody targeting HGF indicate that HGF/c-Met axis activation or inactivation may influence the metabolism of the constituents of fluoropyrimidine-based chemotherapy. The most frequently used fluoropyrimidine as a constituent of fluoropyrimidine-based chemotherapy is 5-fluorouracil (5-FU). Therefore, elucidation of the factors governing the activation of the HGF/c-Met axis and its influence on the metabolism of 5-FU would provide clues for the failure of the aforementioned clinical trials using fluoropyrimidine-based chemotherapy and rilotumumab concomitantly.

It is known that 5-FU exerts its antineoplastic effects through misincorporation of fluoronucleotides into DNA and RNA, and inhibition of nucleotide synthetic enzyme thymidylate synthase (TS) [16]. Furthermore, 5-FU is intracellularly converted to several active metabolites (fluoronucleotides) such as fluorodeoxyuridine monophosphate (FdUMP), fluorodeoxyuridine triphosphate (FdUTP), and fluorouridine triphosphate (FUTP). These conversions are catalyzed by several metabolic enzymes, amongst which, thymidine phosphorylase (TYMP), uridine phosphorylase (UPP), and orotate phosphoribosyl transferase (OPRT) play key roles in the conversion of 5-FU to the initial anabolites and subsequently, to the terminal fluoronucleotides, FdUTP and FUTP, which are misincorporated into DNA and RNA, respectively. Since many metabolic enzymes are involved in the conversion of 5-FU to its active metabolites, the antineoplastic effects of 5-FU are considered to be regulated by several metabolic enzymes influenced by the HGF/c-Met axis activation.

Although 5-FU is a mainstay in the treatment of HCC, monotherapy using 5-FU is limited to the treatment of patients with aberrant HGF/ c-Met axis activation. Agents targeting HGF and c-Met have been vigorously investigated in patients with HCC and human cell lines derived from hepatocyte carcinoma recently. However, despite the fact that additional/synergistic effects are expected to be induced by concomitant use with conventional chemotherapeutic agents, most studies have examined the effects of inhibitors targeting either HGF or c-Met, alone [17,18]. Consequently, influences of HGF on the antineoplastic effects of 5-FU are still poorly understood. Therefore, in the present study we evaluated whether HGF alters the antineoplastic effects of 5-FU in HepG2 cells and explored the possible mechanisms of HGF action. The findings of this study may offer profound insights into the use of chemotherapy comprising inhibitors targeting the HGF/c-Met axis in combination with fluoropyrimidine, in patients with HGF/c-Met axis activation.

## 2. Results

### 2.1. HGF and Epidermal Growth Factor (EGF) Enhances the Antineoplastic Effect of 5-FU

The proliferative activity of HepG2 cells was marginally increased (though not statistically significant) after treatment with either HGF or EGF for 24 h (Figure 1F). We observed that 5-FU decreased the proliferative activity in a concentration dependent manner 36 h after treatment, whereas it enhanced the proliferative activity marginally in the concentration range of 25 to 75 µg/mL, 24 h after treatment. However, this increase in proliferative activity was not statistically significant. In contrast, both HGF and EGF suppressed the proliferative activity of the cells concomitantly treated with 5-FU more potently than that of the cells treated with 5-FU alone at all concentrations 24 h after treatment (Figure 1A–E). Notably, the suppression was more significant at a concentration of 75 µg/mL (Figure 1C); the difference in the suppression of the proliferative activity between the cells concomitantly treated with 5-FU and either HGF or EGF, and those treated with 5-FU alone was 36.74% and 29.78%, respectively.

### 2.2. HGF and EGF Increase UPP1 Expression

It is known that 5-FU exerts antineoplastic effects through its active metabolites produced by conversion by several enzymes (Figure 2A). Therefore, we investigated the effects of HGF and EGF to assess whether the enhancement of antineoplastic effect of 5-FU induced by pretreatment with either of these growth factors results from changes in the expression pattern of genes encoding these enzymes. Although the expression of *UPP2* mRNA was decreased by treatment with either HGF or EGF (Figure 2C), both growth factors significantly increased *UPP1* mRNA expression by 345% and 294%, respectively, 6 h after treatment (Figure 2B). However, the expression of *TYMP* mRNA was marginally increased by HGF and EGF (Figure 2D), and the expression of *OPRT* mRNA was not altered (Figure 2E). Furthermore, we examined the expression of the proteins translated from the respective genes in order to assess whether the observed change in mRNA expression was reflected at the protein level as well. UPP1 protein levels were significantly increased by HGF and EGF to 217% and 184%, respectively, 11 h after treatment (Figure 2F); however, the protein levels of both TYMP and OPRT remained unaltered by treatment with either HGF or EGF (data not shown).

### 2.3. HGF Increases Intracellular Concentration of 5-Fluorouridine (FUR), an Initial Anabolite of 5-FU

UPP1, found to be increased by growth factors in this study, plays an important role in fluoropyrimidine activation. Moreover, UPP1 is the enzyme responsible for the anabolism of 5-FU to FUR, with subsequent phosphorylation to form 5-fluorouridine monophosphate (FUMP) (Figure 2A). Therefore, we measured the concentration of intracellular metabolites of 5-FU after treatment with 5-FU in HepG2 cells pretreated with or without HGF. The concentration of FUR was gradually and significantly increased from 35 min after treatment with 5-FU in the cells treated with HGF (Figure 3B), but remained unaltered in the cells that were not. The concentration of FUMP was marginally increased by pretreatment with HGF; however, this increase was not statistically significant (Figure 3C). Furthermore, the concentration of 5-fluorodeoxyuridine (FUdR), an initial anabolite of 5-FU catalyzed by TYMP, was not altered by pretreatment with HGF (Figure 3D). However, the concentration of FdUMP, a phosphorylated form of FUdR, was apparently increased by pretreatment with HGF (Figure 3E). The concentration of 5-fluorodihydrouracil (DHFU), an inactive metabolite of 5-FU, was not altered (Figure 3F).

### 2.4. Effects of Erlotinib and Gefitinib on HGF-Induced Increase in Expression of UPP1 mRNA Are Antithetical

We investigated the effects of receptor tyrosine kinase (RTK) inhibitors on the growth factor-induced increase in *UPP1* mRNA expression to confirm the involvement of the respective growth factor receptors. The increase in *UPP1* mRNA expression induced by HGF and EGF was suppressed by SU11274 (a MET inhibitor) and erlotinib (an EGFR inhibitor), respectively (Figure 4A). Although the EGF-induced increase was not suppressed by SU11274, the HGF-induced increase was intriguingly enhanced by erlotinib. Therefore, we confirmed concentration-dependence of the enhancement induced by erlotinib and compared it with the effect of gefitinib to validate this unexpected finding. Indeed, erlotinib enhanced the increase induced by HGF in a concentration-dependent manner (Figure 4B). However, gefitinib suppressed the increase induced by HGF, and the intensity of the suppression was attenuated in a concentration-dependent manner (Figure 4C). In contrast, gefitinib consistently suppressed the increase in *UPP1* mRNA expression induced by EGF in a concentration-dependent manner (Figure 4E). However, suppression by erlotinib was also concentration dependent with increased suppression in the concentration range of 0.05 µM to 1 µM; at a higher concentration (5 µM), erlotinib-induced suppression was weaker than that at a concentration of 1 µM (Figure 4D).

### 2.5. Erlotinib Enhances HGF-Induced Increase in Extracellular Signal-Regulated Kinase (Erk) Phosphorylation

To elucidate the mechanisms by which erlotinib enhances the HGF-induced increase in *UPP1* mRNA expression, we investigated the phosphorylation of downstream signaling molecules. Gefitinib suppressed the HGF-induced increase in Erk phosphorylation. However, erlotinib antithetically enhanced the HGF-induced increase in Erk phosphorylation (Figure 5A). In contrast, the EGF-induced increase in Erk phosphorylation was suppressed by both erlotinib and gefitinib (Figure 5B). We have previously reported that signal transduction from Janus Kinase 2 (Jak2) to signal transducer and activator of transcription 3 (STAT3) plays a crucial role in the exertion of HGF effects [15]. In this study, Jak2 and STAT3 were both significantly phosphorylated by HGF, consistent with our previous report. However, contrary to Erk phosphorylation, both erlotinib and gefitinib suppressed the HGF-induced increase in Jak2 and STAT3 phosphorylation (Figure 5C,E). Furthermore, erlotinib and gefitinib both suppressed the EGF-induced increase in Jak2 and STAT3 phosphorylation (Figure 5D,F); however, Jak2 phosphorylation was initially suppressed but was found to be enhanced thereafter (Figure 5D).

### 2.6. Effects of Specific Inhibitors of Downstream Signaling Molecules on HGF- and EGF-Induced Increase in UPP1 mRNA Expression

To further support the results regarding phosphorylation of downstream signaling molecules, we conducted inhibitory experiments using several specific inhibitors. U0126, a mitogen-activated protein kinase (MEK)1/2 inhibitor, completely suppressed the HGF-induced increase in *UPPI* mRNA expression (Figure 6A). In contrast, A6730, an anti-apoptotic serine-threonin kinase 1/2 (AKT1/2) inhibitor, marginally suppressed the HGF-induced increase in *UPPI* mRNA expression, although it was not statistically significant. Stattic, a STAT3 inhibitor, did not suppress the HGF-induced increase in *UPPI* mRNA expression. In the cells treated with EGF, the results using U0126 and Stattic were similar to that in the cells treated with HGF, whereas A6730 marginally enhanced the EGF-induced increase in *UPPI* mRNA expression, although it was not statistically significant (Figure 6B). Conversely, the HGF- and EGF-induced increase in *TYMP* mRNA were both suppressed by all the inhibitors tested (Figure 6C,D). These results indicate that mitogen-activated protein kinase (MAPK)/Erk pathway is mainly involved in growth factor-induced increase in *UPP1* mRNA expression.

### 2.7. Erlotinib Markedly Suppresses the HGF-Induced Increase in EGFR Phosphorylation at the Tyr1173 Sight

Western blotting was performed to elucidate the phosphorylation sites of EGFR responsible for the enhancement of HGF-induced increase in *UPP1* mRNA expression in the cells treated with erlotinib. In the cells treated with HGF (Figure 7A,C,E), increased phosphorylation of EGFR at the Tyr1173 site was markedly suppressed by pretreatment with erlotinib (Figure 7E). However, gefitinib modestly suppressed the increased phosphorylation. In contrast, although erlotinib did not suppress the increased phosphorylation of EGFR at the Tyr845 and Tyr1068 sites (Figure 7A,C), gefitinib did not suppress the increased phosphorylation at the Tyr845 site (Figure 7A), but enhanced phosphorylation at the Tyr1068 site (Figure 7C). In the cells treated with EGF (Figure 7B,D,F), erlotinib significantly suppressed phosphorylation at the Tyr845 and Tyr1173 sites (Figure 7B,F). In contrast, erlotinib and gefitinib both marginally augmented phosphorylation at the Tyr1068 site (Figure 7D).

### 2.8. Effects of Receptor Tyrosine Kinase Inhibitors on Growth Factor-Induced Phosphorylation of c-Met

We investigated the effects of receptor tyrosine kinase inhibitors on the phosphorylation of c-Met to confirm whether the enhancement of HGF-induced increase in *UPP1* mRNA expression in the cells treated with erlotinib is induced via c-Met. HGF significantly increased in phosphorylation at the Tyr1003 site, and the increase was significantly suppressed by both erlotinib and gefitinib (Figure 8A); however, phosphorylation at the Tyr1349 site was significantly enhanced by erlotinib, and marginally and transiently enhanced by gefitinib (Figure 8E). In contrast, EGF-induced increase in phosphorylation at all observed phosphorylation sites were suppressed by both erlotinib and gefitinib (Figure 8B,D,F).

## 3. Discussion

In the present study, we demonstrated that HGF increases UPP1 expression, resulting in enhancement of antineoplastic effects of 5-FU. To the best of our knowledge, this is the first study to report a positive role of HGF in augmenting the effects of an anti-cancer agent. Furthermore, we found that HGF-induced activation of the c-Met signaling pathway molecule, Erk involved in UPP1 expression, was enhanced by an EGFR TKI, erlotinib. These findings suggest that chemotherapy using 5-FU is likely to be more effective in patients with activation of the HGF/c-Met axis compared to that in those without. Therefore, Concomitant treatment with an inhibitor targeting the HGF/c-Met axis suppresses the anticancer effects of 5-FU in patients with activation of the HGF/c-Met axis. In contrast, inhibition of EGFR induced by erlotinib enhances the anticancer effects of 5-FU.

5-FU is widely used in the treatment of multiple cancers such as liver, lung, colon, breast, pancreatic, ovarian, and gastric carcinomas. In addition, aberrant activation of the HGF/c-Met axis has been reported in aforementioned cancer types [4,5,6], and is considered to be closely associated with the resistance to chemotherapeutic agents [18,19]. Therefore, combination of fluoropyrimidine with an inhibitor of the HGF/c-Met axis has been expected to overcome the resistance. However, concomitant administration of rilotumumab, an antibody targeting HGF, with a fluoropyrimidine-based chemotherapy regimen has been recently reported to cause disease progression, resulting in deaths in c-Met–positive patients [12,13]. Therefore, here, we investigated the effects of HGF on HepG2 cells treated with 5-FU. Although the proliferative activity of the cells treated with HGF alone was marginally enhanced 24 h after treatment (Figure 1F), that of the cells treated with 5-FU was more potently suppressed by pretreatment with HGF (Figure 1A–E). In the concentration range of 25 µg/mL to 75 µg/mL, 5-FU alone marginally enhanced the proliferative activity 24 h after treatment (Figure 1A–C); this may be explained by the fact that uracil supplied by 5-FU was used as a substrate in the enzymatic synthesis of DNA and RNA, and the cytotoxic effect caused by the misincorporation of 5-FU metabolites into the nucleic acids was not evident initially. On the other hand, it is necessary to examine whether concurrent administration of anti-HGF monoclonal antibody and 5-FU directly enhances toxicity, since it has been reported in recent years that the number of deaths associated with disease progression increases when anti-HGF monoclonal antibody is combined with fluoropyrimidine chemotherapy [12]. In this regard, we speculate that the toxicity of 5-FU to normal cells may be enhanced by anti-HGF monoclonal antibodies. Furthermore, combined effects of anti-HGF monoclonal antibody and 5-FU in the presence of HGF should also be examined. However, since HGF is neutralized by anti-HGF monoclonal antibodies in the culture medium, anti-HGF monoclonal antibodies may attenuate the HGF-induced enhancement of the antiproliferative effect of 5-FU. However, these points require further investigation.

The antineoplastic effects of 5-FU are exerted via its active metabolites (Figure 2A). Conversion of 5-FU to these metabolites is catalyzed by several enzymes. Therefore, in the present study, gene expression of the enzymes involved in the conversion process was investigated to evaluate the enzymes responsible for augmenting the antineoplastic effects of 5-FU induced by pretreatment with HGF. Among the enzymes (OPRT, TYMP, and UPP1) involved in the conversion of 5-FU to the initial anabolites, *UPP1* mRNA expression was markedly increased by HGF, being more than three-fold of that in the control cells 6 h after treatment (Figure 2B). Similarly, *UPP1* mRNA expression increased nearly three-fold following treatment with EGF. Furthermore, consistent with the gene expression results, UPP1 protein expression levels were significantly increased both by HGF and EGF (Figure 2F). Increased UPP1 has been reported to enhance the antineoplastic effects of 5-FU [20,21]. On the basis of these reports and our results, we hypothesized that HGF enhances the antineoplastic effect of 5-FU through the upregulation of UPP1 expression. Therefore, we next examined the intracellular concentration of 5-FU and its anabolites converted by the aforementioned enzymes in the cells pretreated with HGF. FUR, an initial anabolite converted by UPP1, was significantly increased by pretreatment with HGF (Figure 3B). In contrast, FUMP, an initial anabolite converted by OPRT, was marginally increased (Figure 3C), consistent with the mRNA expression profile of OPRT, which was not increased by HGF (Figure 2E). Furthermore, FUdR, an initial anabolite converted by TYMP, was not altered by HGF (Figure 3D). These findings strongly support our hypothesis. Nevertheless, because FdUMP, a phosphorylated form of FUdR, was increased in the cells treated with HGF (Figure 3E), TYMP which catalyzes the conversion of 5-FU to FUdR may be implicated in the HGF-enhanced antineoplastic effects of 5-FU. Although *TYMP* mRNA, which is translated into TYMP, was slightly increased (Figure 2D), it was transient and insufficient to increase FUdR concentration. Therefore, the increase in intracellular FdUMP may be explained by the fact that FUR was increased by an up-regulated UPP1 and was sequentially converted as follows: FUR → FUMP → fluorouridine diphosphate (FUDP) → fluorodeoxyuridine diphosphate (FdUDP) → FdUMP. To evaluate this consideration, we further investigated the mRNA expression of enzymes involved in these conversions. Although mRNA expression of uridine kinase (UK), an enzyme which catalyzes the conversion of FUR to FUMP, was not altered by treatment with HGF (Appendix A), that of ribonucleotide reductase subunit M2 (RRM2), a subunit of ribonucleotide reductase which catalyzes the conversion of FUDP to FdUDP, was modestly increased (Appendix A). These results indicate that the conversion of FUDP to FdUDP, and subsequently to FdUMP, are modestly increased. RRM2 has been reported to play a promotive role in cancer progression due to producing excessive deoxynucleotide triphosphates (dNTPs) [22,23]. Conversely, RRM2 plays a positive role in the expression of antineoplastic effects of 5-FU due to conversion of 5-FU to its active metabolites FdUMP and FdUTP. Therefore, it is plausible that HGF enhances the antineoplastic effects of 5-FU by increasing UPP1 expression.

Since c-Met has been reported to interact with EGFR [24], we examined whether increased UPP1 expression induced by HGF was through its own receptor c-Met alone. Therefore, we investigated the effects of RTK inhibitors on the growth factor-induced increase in *UPP1* mRNA. Indeed, the HGF-induced increase in *UPP1* mRNA expression was significantly inhibited by pretreatment with SU11274, a c-Met inhibitor (Figure 4A). Intriguingly, although erlotinib is an EGFR inhibitor, it significantly enhanced *UPP1* mRNA transcription induced by HGF in a concentration-dependent manner (Figure 4B). In contrast, gefitinib, which is also an EGFR inhibitor, did not enhance *UPP1* mRNA transcription, but on the contrary inhibited it (Figure 4C). c-Met and EGFR have been shown to share overlapping downstream signaling pathways such as MAPK and phosphatidylinositol-3-kinase (PI3K)/AKT and can trans-phosphorylate one another [25]. We have previously reported that HGF transactivates EGFR and enhances phosphorylation of downstream signaling molecules Jak2/STAT3 in HepG2 cells [15]. Therefore, to elucidate the mechanisms of the antithetical effects of erlotinib and gefitinib, we explored the differences in phosphorylation of their downstream signaling molecules. HGF- and EGF-induced increase in phosphorylation of Jak2 and STAT3 were suppressed by both erlotinib and gefitinib (Figure 5C–F). In contrast, although the EGF-induced increase in phosphorylation of Erk was suppressed by both erlotinib and gefitinib (Figure 5B), the HGF-induced increase in phosphorylation of Erk was enhanced by erlotinib, but suppressed by gefitinib (Figure 5A). We further evaluated the influences of Akt; EGF-induced increase in phosphorylation of Akt was suppressed by both erlotinib and gefitinib (Appendix A). In contrast, although the HGF-induced increase in phosphorylation of Akt was marginally enhanced by erlotinib, the magnitude of enhancement was insufficient to further enhance the increase in *UPP1* mRNA induced by HGF; gefitinib did not suppress the HGF-induced increase in phosphorylation (Appendix A). These results indicated that Jak2/STAT3 and Akt are not associated with the growth factor-induced increase in *UPP1* mRNA. Furthermore, we investigated the phosphorylation sites of EGFR responsible for the enhancement of Erk phosphorylation induced by erlotinib. HGF strongly increased EGFR phosphorylation at the Tyr1173 site, and the increase was markedly suppressed by erlotinib (Figure 7E). In contrast, gefitinib modestly suppressed the increase. The phosphorylation site at Tyr1173 is located at a multifunctional docking site, and plays an important role in the activation of subsequent downstream signaling molecules. Phospho-Tyr1173 has been reported to play two opposite roles in EGFR signaling and cooperate with other phosphorylation sites to elicit Erk signaling through recruitment of Src homology 2 (SH2)-domain-containing transforming protein (SHC) and growth factor receptor-bound protein 2 (Grb2) [26,27]. Conversely, it works alone to attenuate Erk activation through SH2-domain-containing protein tyrosine phosphatase 1 (SHP1) binding [28]. Furthermore, Hsu et al., showed that downregulation of phosphorylation at the Tyr1173 site diminishes the recruitment of SHP1, but not that of SHC or Grb2 [29]. Therefore, erlotinib-induced enhancement of *UPP1* mRNA expression induced by HGF may be attributed to Erk activation through suppression of the HGF-induced increase in phosphorylation at the Tyr1173 site. In contrast, gefitinib-induced suppression of phosphorylation at the Tyr1173 site is considered to be insufficient to diminish the recruitment of SHP1, resulting in the suppression of Erk phosphorylation due to inhibitory effects of gefitinib on other phosphorylation sites.

Nevertheless, although erlotinib is an EGFR inhibitor and EGFR interacts with c-Met, it cannot be completely ruled out that c-Met is involved in the erlotinib-induced enhancement. Therefore, we examined the phosphorylation of c-Met. EGF-induced increase in phosphorylation at all observed phosphorylation sites were suppressed by both erlotinib and gefitinib (Figure 8B,D,F). In contrast, the HGF-induced increase in phosphorylation at the Tyr1003 site was significantly suppressed by both erlotinib and gefitinib (Figure 8A); however, phosphorylation at the Tyr1349 site was significantly enhanced by erlotinib, and marginally and transiently enhanced by gefitinib (Figure 8E). The Tyr1003 site, which is located at the juxtamembrane domain, plays a role in the negative regulation of c-Met [30,31]. Therefore, no involvement of the Tyr1003 site is considered in the erlotinib-induced enhancement of *UPP1* mRNA expression. Although Tyr1349 is located at a multifunctional docking site and is involved in Erk activation [32,33], it may not be directly involved in the erlotinib-induced enhancement of *UPP1* mRNA expression. This may be explained by the fact that the HGF-induced enhancement of phosphorylation as observed in this study at this site might be insufficient to augment Erk activation. Furthermore, HGF-induced increase in *UPP1* mRNA expression was completely suppressed by MEK 1/2 inhibitor U0126 (Figure 6A). Therefore, these results strongly support the possibility that erlotinib-induced enhancement of the HGF-induced increase in *UPP1* mRNA expression is caused by Erk activation through erlotinib-induced suppression of HGF-enhanced EGFR phosphorylation at the Tyr1173 site.

In conclusion, Erk activation is induced by HGF directly, enhancing tumor proliferation; however, the enhancement of proliferation is considered to accelerate the misincorporation of 5-FU metabolites into DNA and RNA, which is further augmented by UPP1 overexpression, induced by HGF. Consequently, the antineoplastic effect of 5-FU is synergistically enhanced. Therefore, our findings imply that concomitant treatment with fluoropyrimidine-based chemotherapy and an inhibitor targeting the HGF/c-Met axis will diminish the antineoplastic effects of the treatment, resulting in its failure such as that previously reported in clinical trials [12,13]. In addition, erlotinib increases the effect of fluoropyrimidine-based chemotherapy. Therefore, the assessment of the HGF/c-Met axis status prior to addition of its inhibitor to a fluoropyrimidine-based chemotherapy regimen is critical for safe and successful treatment. Moreover, among RTK inhibitors targeting EGFR, erlotinib should be selected in the concomitant treatment with fluoropyrimidine-based chemotherapy. However, further prospective clinical investigations are needed to confirm the negative effect of inhibitors targeting the HGF/c-Met axis in the concomitant treatment with fluoropyrimidine-based chemotherapy and to verify the benefits of erlotinib in this therapeutic regimen.

## 4. Materials and Methods

### 4.1. Materials

Growth factors (HGF and EGF) and tyrosine kinase inhibitors (gefitinib and erlotinib) were purchased from PeproTech Inc. (Rocky Hill, CT, USA) and Wako Pure Chemical Industries Ltd. (Osaka, Japan), respectively. A6730, 5-FU, SU11274, U0126, and Stattic were purchased from Sigma-Aldrich Co. (St. Louis, MO, USA). Antibodies were obtained from the following sources and used at the indicated dilutions: Sigma-Aldrich Co. (St. Louis, MO, USA), UPP1 (1:600), p-STAT3, and α-tubulin (1:10,000); Cell Signaling Technology, Inc. (Danvers, MA, USA), p-c-Met Tyr1003, Tyr1234/1235, Tyr1349, p-EGFR Tyr845, Tyr1068, Tyr1173, p-Akt1/2, and p-Jak2 (all 1:10,000); Santa Cruz Biotechnology (Dallas, TX, USA), p-Erk1/2 (1:2000). Other reagents and biochemicals were purchased from Sigma-Aldrich Co. (St. Louis, MO, USA).

### 4.2. Cell Culture

The proliferative effects of HGF on cancer cells could mask its effects on the metabolism of 5-FU. Since HepG2 cells hardly proliferate in response to HGF (Figure 1F), these cells were selected as our experimental model. HepG2 cells procured from RIKEN BioResource Center (Ibaraki, Japan) were pre-cultured in Minimum Essential Medium supplemented with 10% fetal bovine serum and antibiotics (100 U/mL penicillin and 100 µg/mL streptomycin). The medium was replaced with fresh serum-free medium and cultured with or without growth factors (HGF and/or EGF: 50 ng/mL) at 37 °C in 5% CO_2_.

### 4.3. Cell Proliferation Assay

HepG2 cells were plated in 96-well plates at 3 × 10^3^ cells/well, grown in medium supplemented with 10% fetal bovine serum for 24 h, and were continuously cultured in serum-free medium with or without HGF or EGF for 7 h. Thereafter, the cells were treated with 5-FU for the indicated times. Proliferative activity was detected using the CellTiter 96 Aqueous One Solution Cell Proliferation Assay Kit (Promega Corporation, Madison, WI, USA).

### 4.4. Real-Time Reverse Transcription-Polymerase Chain Reaction (RT-PCR)

Total RNA was purified from the HepG2 cells using a commercial kit (RNeasy Mini Kit; Qiagen Inc., Valencia, CA, USA). The amount of total RNA extracted from HepG2 cells was quantified using a NANODROP LITE (Thermo Fisher Scientific, Waltham, MA, USA). A reverse transcription reaction was performed using the PrimeScript RT reagent kit (Takara Bio, Otsu Japan). cDNA synthesized from 100 ng of total RNA was added to a PCR mixture containing SYBR Premix Ex Taq II (Takara Bio, Otsu Japan), one of the primer sets (OPRT [forward primer: 5’-TTG AAG ACC GGA AGT TTG CAG ATA-3’; reverse primer: 5’-CTG GCA CCA CGT GAG CAT TTA-3’], TYMP [forward primer: 5’-GGC TGC TGT ATC GTG GGT CA-3’; reverse primer: 5’-GAA CTT AAC GTC CAC CAC CAG AG -3’], UPP1 [forward primer: 5’-GGT GCT CCA ACG TCA CTA TCA TC-3’; reverse primer: 5’-TGC AGA ACA CAG CAA CAG CTC-3’], and UPP2 [forward primer: 5’-ATG GAA TCT ACA GTG TTT GCA GCT A-3’; reverse primer: 5’-TGG TAC TCC ACC AGG ACA TCA-3’]), and RNase-free distilled water. PCR was performed using the Thermal Cycler Dice Real-Time System (Takara Bio, Otsu, Japan), and the cycling conditions were as follows: incubation for 30 s at 95 °C, 45 cycles of 3 s each at 60 °C, and 30 s at 95 °C.

### 4.5. Western Blot Analysis

HepG2 cells, cultured in 6-well plates with or without HGF or EGF for the indicated periods, were lysed in 140 µL of RIPA buffer (Sigma-Aldrich, St. Louis, MO, USA). The lysate was centrifuged at 20,000× *g* for 10 min at 4 °C. Equal amounts of protein extract were separated by electrophoresis using 7–10% polyacrylamide gels containing sodium dodecyl sulfate and were transferred to polyvinyl difluoride membranes. The membranes were then blocked with a blocking solution (5% skim milk or 1% BSA) for 1 h at 23 °C and were incubated overnight at 4 °C with primary antibodies. After thorough washing, the membranes were incubated with a secondary antibody for 1 h at 23 °C. The labeled blots were visualized using an enhanced chemiluminescence detection system (ECL Prime; GE Healthcare, Bucks, UK).

### 4.6. Measurement of 5-FU and Its Metabolites in HepG2 Cells

HepG2 cells were cultured in 24-well plates with or without HGF for 7 h. Thereafter, 5-FU was added to the medium and incubated for the indicated times. At the end of the incubation period, the cells were washed with ice-cold phosphate buffered saline, and 250 µL of the extraction solution composed of 0.5 N HCL was added. The cells were incubated at 4 °C overnight for complete lysis. The extracts were neutralized by adding 250 µL of 0.5 N NaOH. In order to determine the protein concentration, 100 µL aliquots of the extracts were used. The remaining extracts were centrifuged, and the supernatants were collected as intracellular samples. The samples were extracted according to previously described methods with minor modifications [34]. A 50 µL aliquot of the supernatant was injected into a high-performance liquid chromatography system (Prominence UFLC; Shimadzu Co., Kyoto, Japan) equipped with a Shim-pack XR ODSII column (3.0 mm × 150 mm; Shimadzu Co., Kyoto, Japan). The column effluent was monitored by a UV detector set at 266 or 215 nm.

### 4.7. Statistical Analysis

Data are expressed as mean ± SD. Differences among groups were assessed using Dunnett’s multiple comparisons test or unpaired Student’s *t*-test after analysis of variance (ANOVA) was performed. Statistically significant differences are indicated by *p* < 0.05 and *p* < 0.01.

## Figures and Tables

**Figure 1 ijms-23-09108-f001:**
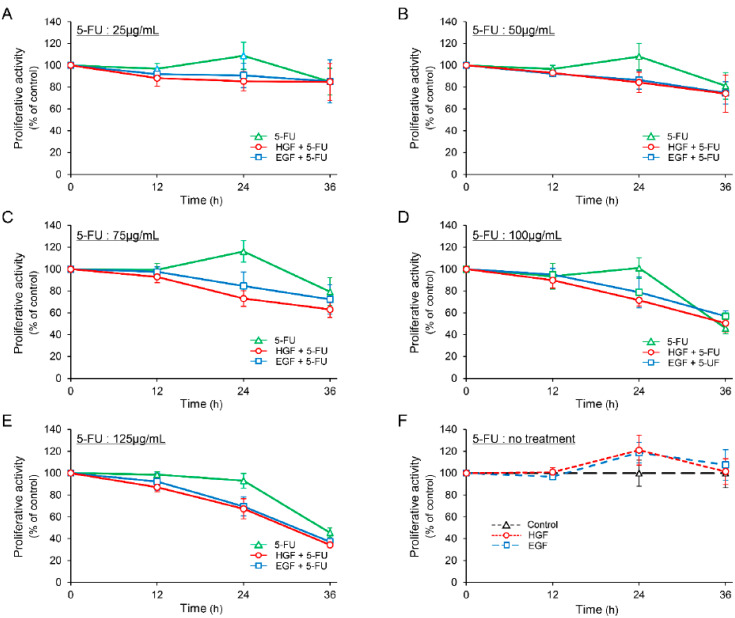
Hepatocyte growth factor (HGF) and epidermal growth factor (EGF) suppress proliferative activity in HepG2 cells treated with 5-fluorouracil (5-FU). HepG2 cells were cultured for 24 h in serum-starved medium. Incubation was continued with addition of either HGF (50 ng/mL) or EGF (50 ng/mL) for further 7 h. Thereafter, 5-FU (25 µg/mL: (**A**) 50 µg/mL: (**B**) 75 µg/mL: (**C**) 100 µg/mL: (**D**) 125 µg/mL: (**E**)) and vehicle (**F**) were added. The proliferative activity was measured at the indicated times and is expressed as a percentage of the proliferative activity of cells not treated with growth factor and 5-FU at 0 h (**A**–**E**) or at the corresponding times (**F**). Each bar represents the mean ± SD of 5 experiments.

**Figure 2 ijms-23-09108-f002:**
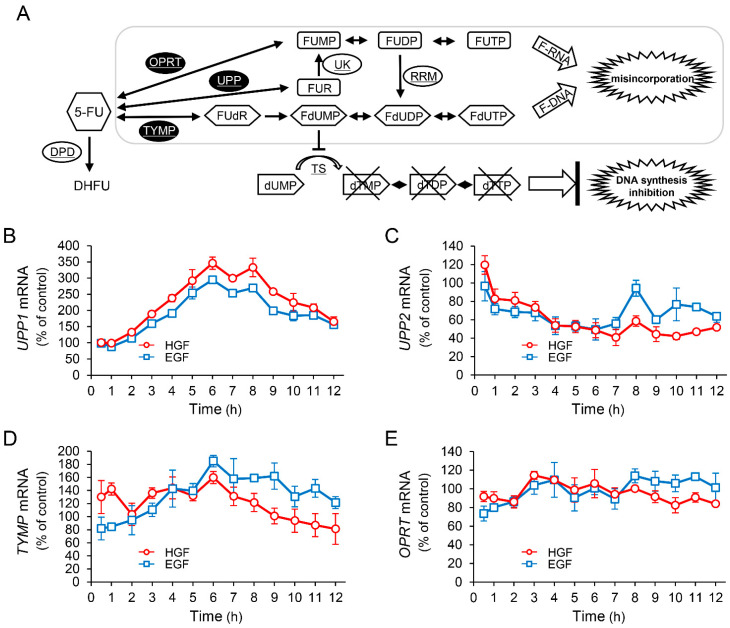
HGF and EGF increase expression of uridine phosphorylase 1 (*UPP1)* mRNA and its protein. Metabolism of 5-FU is shown in panel (**A**). DPD, dihydropyrimidine dehydrogenase; OPRT, orotate phosphoribosyl transferase; RRM, ribonucleotide reductase subunit M; TS, thymidylate synthase; TYMP, thymidine phosphorylase; UK, uridine kinase; UPP, uridine phosphorylase; 5-FU, 5-fluorouracil; DHFU, fluorodihydrouracil; dUMP, deoxyuridine monophosphate; dTMP, deoxythymidine monophosphate; dTDP, deoxythymidine diphosphate; dTTP, deoxythymidine triphosphate; FUMP, fluorouridine monophosphate; FUDP, fluorouridine diphosphate; FUTP, fluorouridine triphosphate; FUR, fluorouridine; FUdR, fluorodeoxyuridine; FdUMP, fluorodeoxyuridine monophosphate; FdUDP, fluorodeoxyuridine diphosphate; FdUTP, fluorodeoxyuridine triphosphate. HepG2 cells were incubated with either HGF (50 ng/mL) or EGF (50 ng/mL) for the indicated period. Thereafter, total RNA and protein were extracted. The transcription levels of *UPP1* (**B**), *UPP2* (**C**), *TYMP* (**D**), and *OPRT* (**E**) are expressed as a percentage of that in cells not treated with growth factors at 0 h. Each bar represents the mean ± SD of 5 experiments. Total protein was collected at the indicated time points. The expression level of the UPP1 protein was detected by western blotting. Western blot analysis was performed in 3 experiments for 3 different preparations, and the representative blots are shown. The density of the bands was measured using ImageQuant TL software (Ver. 7.0, EG Healthcare Inc., Chicago, IL, USA). The levels of protein expression quantified by scanning densitometry of the immunopositive protein and corrected for α-tubulin levels in the same samples are represented as a percentage of that in cells not treated with growth factors at 0 h (**F**).

**Figure 3 ijms-23-09108-f003:**
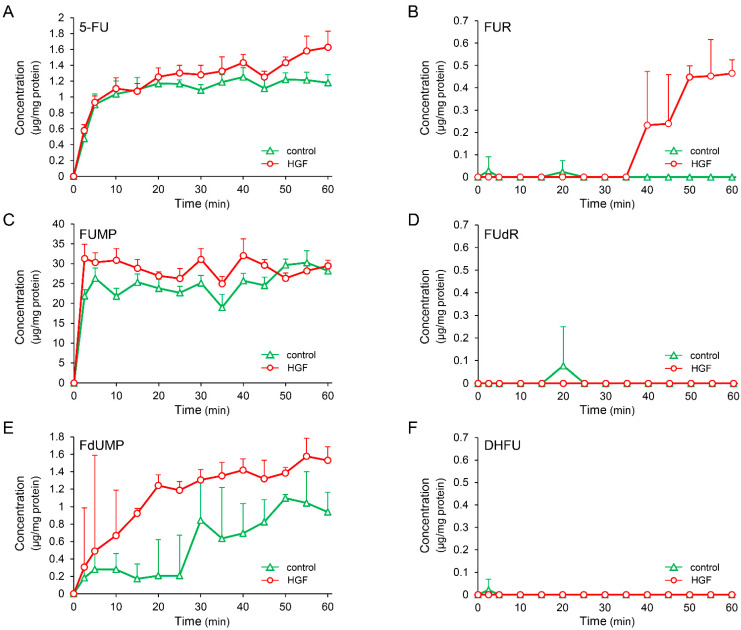
HGF increases concentration of FUR, an initial anabolite of 5-FU converted by UPP1. HepG2 cells, with or without HGF (50 ng/mL) treatment, were cultured for 7 h, followed by addition of 5-FU. The anabolites were collected at the indicated time points, and the concentration was determined by dividing the amount of anabolite by the protein concentration of the cell lysate. Each bar represents the mean ± SD of 5 experiments. 5-FU: (**A**), FUR: (**B**), FUMP: (**C**), FUdR: (**D**), FdUMP: (**E**), DHFU: (**F**).

**Figure 4 ijms-23-09108-f004:**
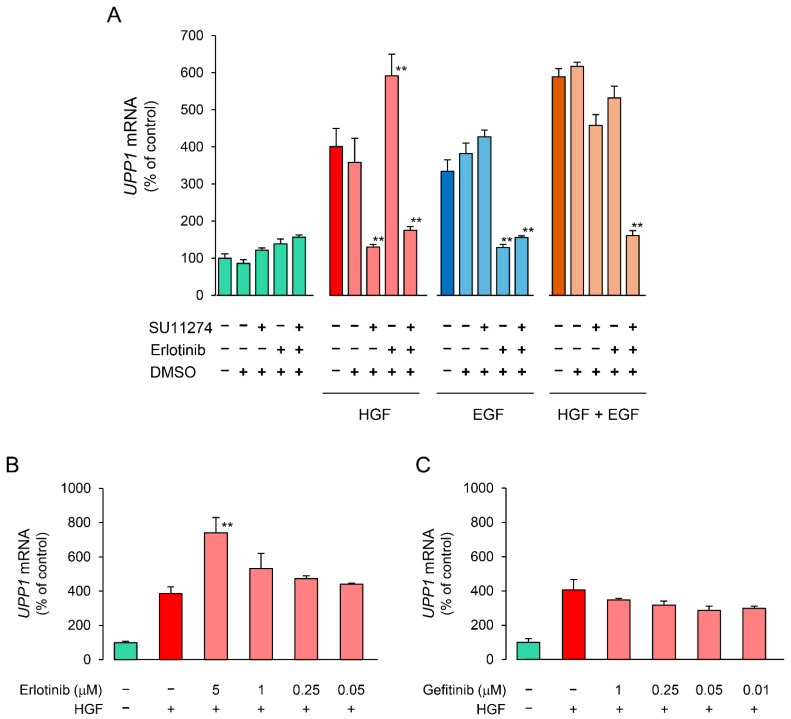
Inhibitory effects of receptor tyrosine kinase (RTK) inhibitors on HGF- and EGF-induced enhancement of *UPP1* mRNA expression. HepG2 cells were pretreated with SU11274 (5 µM), a MET inhibitor, erlotinib ((**A**) 5 µM; B-E, indicated concentration), and gefitinib ((**B**–**E**) indicated concentration) 15 min before administration of growth factors, and were continued to be cultured for 7 h after treatment with HGF (50 ng/mL), EGF (50 ng/mL) or both. DMSO, used as vehicle, has no effect on the expression of mRNA. At the end of the incubation period, the expression level of mRNA was measured and was expressed as a percentage of that in cells not treated with growth factors and RTK inhibitors. Each bar represents the mean ± SD of 5 experiments. Significantly different from the level in cells treated with growth factor alone: ** *p* < 0.01, * *p* < 0.05.

**Figure 5 ijms-23-09108-f005:**
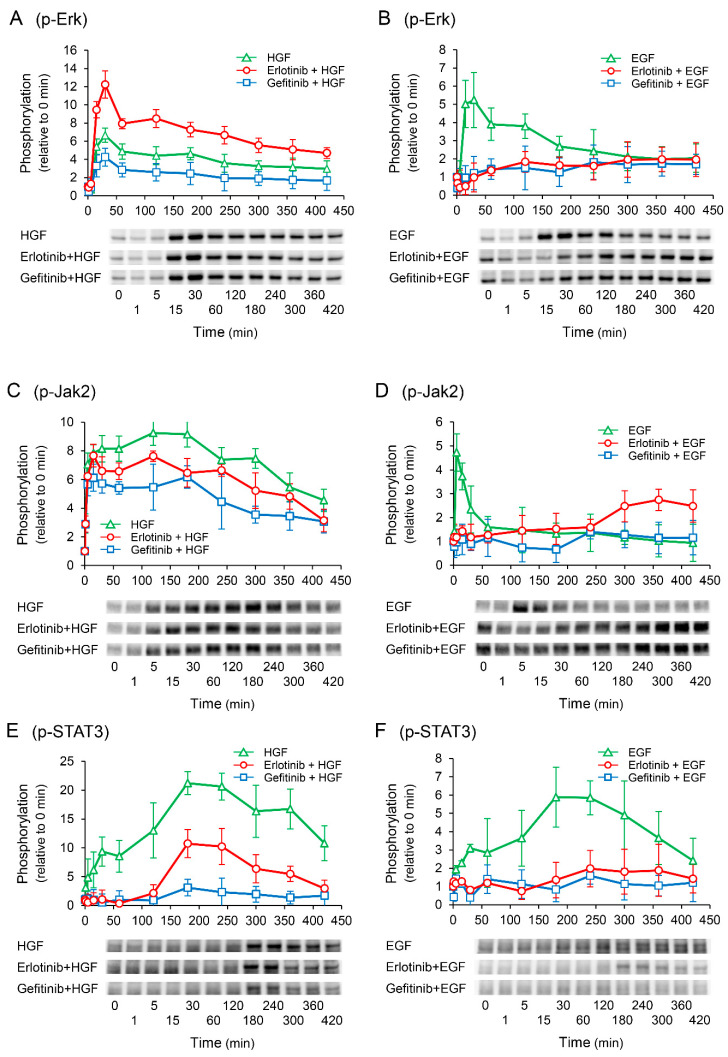
Effects of erlotinib and gefitinib on growth factor-induced phosphorylation of downstream signaling molecules. HepG2 cells were cultured for 24 h in serum-starved medium. Thereafter, HGF (50 ng/mL) or EGF (50 ng/mL) was added to the medium. Cells were pretreated with erlotinib (5 µM) or gefitinib (1 µM) 15 min before treatment with growth factors. Total protein was collected at the indicated time points. The relative amount of phosphorylated protein in each band was quantified by ImageQuant TL software, and phosphorylation trends of extracellular signal-regulated kinase (Erk) (**A**,**B**), Janus Kinase 2 (Jak2) (**C**,**D**), and signal transducer and activator of transcription 3 (STAT3) (**E**,**F**) after treatment with growth factors were plotted on graphs. The intensity of phosphorylation is expressed relative to that at 0 min. Western blot analysis was performed in 3 experiments for 3 different preparations, and representative blots are shown. Each bar represents the mean ± SD.

**Figure 6 ijms-23-09108-f006:**
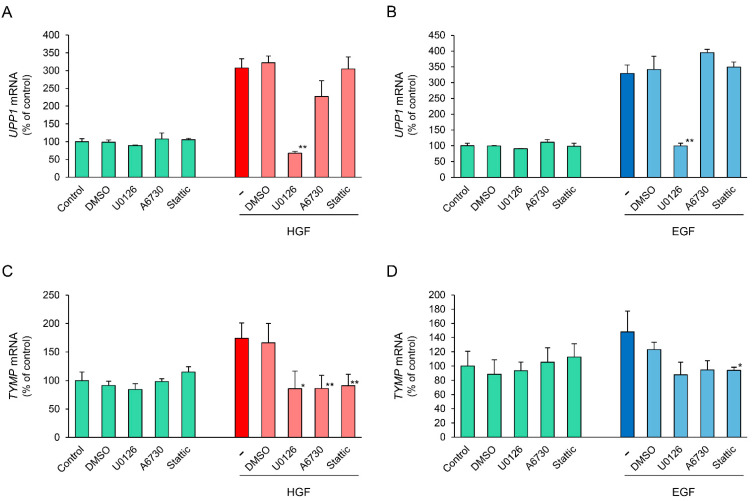
Comparison of inhibitory effects of inhibitors of downstream signaling molecules on HGF- and EGF-induced increase in *UPP1* and *TYMP* mRNA expression. HepG2 cells were pretreated with U0126 (2.5 µM), a mitogen-activated protein kinase (MEK1/2) inhibitor, A6730 (5 µM), an anti-apoptotic serine-threonin kinase 1/2 (Akt1/2) inhibitor, and Stattic (10 µM), a STAT3 inhibitor 15 min before administration of growth factors. The inhibitors were used at the concentration at which they produced no effect on the spontaneous expression of mRNA. The expression level of *UPP1* mRNA in cells treated with HGF (50 ng/mL) (**A**) or EGF (50 ng/mL) (**B**) and the expression level of *TYMP* mRNA in cells treated with HGF (**C**) or EGF (**D**) was measured 6 h after treatment with growth factors, and was expressed as a percentage of that in cells not treated with inhibitors and growth factors. Each bar represents the mean ± SD of 5 experiments. Significantly different from the level in cells treated with growth factor alone: ** *p* < 0.01, * *p* < 0.05.

**Figure 7 ijms-23-09108-f007:**
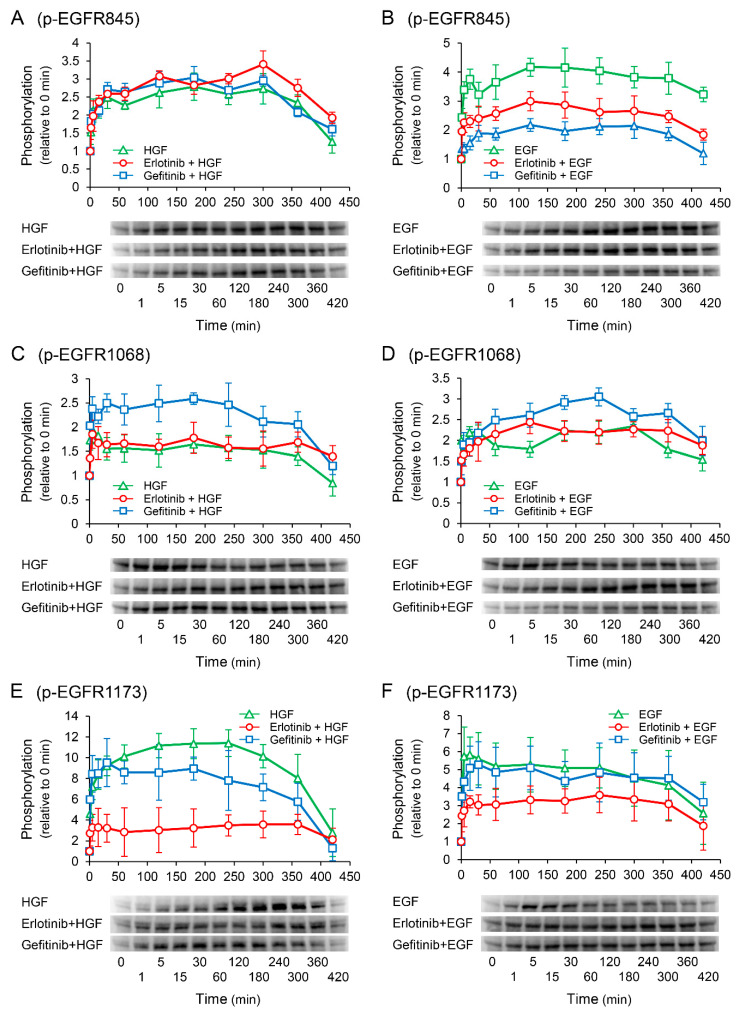
Effects of erlotinib and gefitinib on growth factor-induced phosphorylation of epidermal growth factor receptor (EGFR). HepG2 cells were cultured for 24 h in serum-starved medium. Thereafter, HGF (50 ng/mL) or EGF (50 ng/mL) was added to the medium. Cells were pretreated with erlotinib (5 µM) or gefitinib (1 µM) 15 min before treatment with growth factors. Total protein was collected at the indicated times points. The relative amount of phosphorylated protein in each band was quantified by ImageQuant TL software, and phosphorylation trends at Tyr845 (**A**,**B**), Tyr1068 (**C**,**D**), and Tyr1173 (**E**,**F**) after treatment with growth factors were plotted on graphs. The intensity of phosphorylation is expressed relative to that at 0 min. Western blot analysis was performed in 3 experiments for 3 different preparations, and representative blots are shown. Each bar represents the mean ± SD.

**Figure 8 ijms-23-09108-f008:**
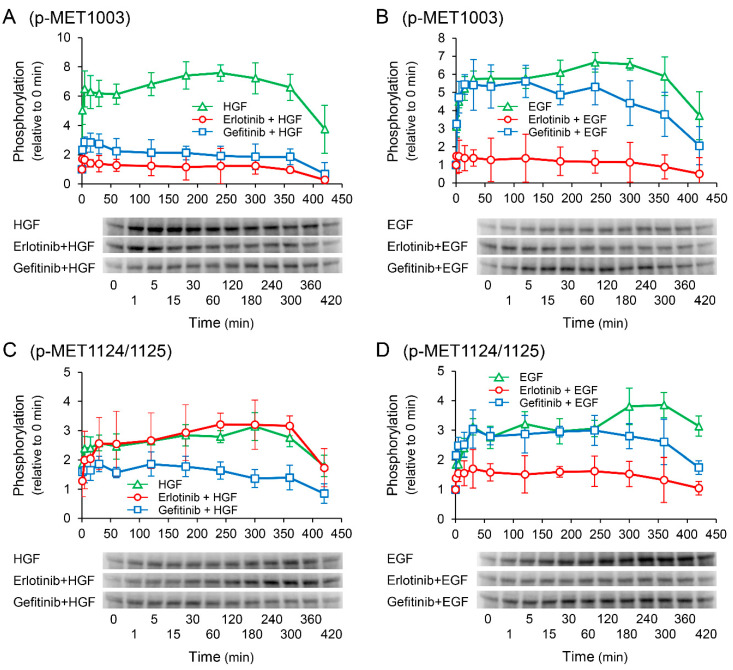
Effects of erlotinib and gefitinib on growth factor-induced phosphorylation of c-Met. HepG2 cells were cultured for 24 h in serum-starved medium. Thereafter, HGF (50 ng/mL) or EGF (50 ng/mL) was added to the medium. Cells were pretreated with erlotinib (5 µM) or gefitinib (1 µM) 15 min before treatment with growth factors. Total protein was collected at the indicated time points. The relative amount of phosphorylated protein in each band was quantified by ImageQuant TL software, and phosphorylation trends at Tyr1003 (**A**,**B**), Tyr1124/1125 (**C**,**D**), and Tyr1349 (**E**,**F**) after treatment with growth factors were plotted on graphs. The intensity of phosphorylation is expressed relative to that at 0 min. Western blot analysis was performed in 3 experiments for 3 different preparations, and representative blots are shown. Each bar represents the mean ± SD.

## Data Availability

The data supporting the finding of this study are contained within the contents of this article. The dataset generated during this study will be freely provided by the corresponding author upon request.

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
