# Peer review of "Hepatocyte Growth Factor Enhances Antineoplastic Effect of 5-Fluorouracil by Increasing UPP1 Expression in HepG2 Cells"

_ijms, 2022, doi:10.3390/ijms23169108_

Round 1

Reviewer 1 Report

The study presented by Okumura et al. aims at investigating the influence of HGF treatment on the antineoplastic effects of 5-FU. The authors investigated gene and protein expression of enzymes involved in the metabolism of 5-FU and showed that HGF treatment increased significantly the expression of UPP1 resulting in enhancement of the antineoplastic effect of 5-FU. The authors imply that concomitant treatment of patients with 5-FU and anti HGF/c-Met inhibitors would diminish the antineoplastic effects of the treatments, resulting in a failure.

The manuscript is well written and the experiments well performed. However, I have the following issues that need to be considered:

1.       In your study you did not use anti-HGF inhibitors (monocolonal antibodies). How would these influence on the expression of UPP1?

2.       The enhanced toxicity of anti-HGF + 5-FU is not experimentally explained/discussed.

Author Response

August 1, 2022

Dear reviewer:

We are most grateful to you for the critical comments and useful suggestions on the original version of our manuscript entitled “Hepatocyte growth factor enhances antineoplastic effect of 5-fluorouracil by increasing UPP1 expression in HepG2 cells” (manuscript no.: ijms-1815911). These comments and suggestions have helped us to improve our manuscript considerably.

As indicated in the responses that follow, we have carefully considered the comments and suggestions made by you in the revised version of our manuscript.

Response to Reviewer 1 comments

Point 1: In your study you did not use anti-HGF inhibitors (monoclonal antibodies). How would these influence on the expression of UPP1?

Response 1: We are also interested in the effect of anti-HGF monoclonal antibodies on the expression of UPP1. However, we have not experimented with anti-HGF monoclonal antibodies. It is unclear whether anti-HGF monoclonal antibodies directly affect the expression of UPP1. If we evaluated the effects of anti-HGF monoclonal antibodies in our experimental design, it is considered that the upregulation of UPP1 expression by HGF was suppressed as in the case of receptor antagonists. I would like to evaluate it in future research. Thank you for your valuable comment.

Point 2: The enhanced toxicity of anti-HGF + 5-FU is not experimentally explained/discussed.

Response 2: As indicated by the reviewer, we should have considered whether the antiproliferative effect of 5-FU is enhanced by simultaneous administration of anti-HGF monoclonal antibodies. In the culture environment, HGF is neutralized by additionally administered anti-HGF monoclonal antibodies, and it is speculated that only the antiproliferative effect of 5-FU is exhibited. However, we have not experimented with anti-HGF monoclonal antibodies in this study. Therefore, we have retouched the discussion to explain on this point (Please see page 13, line 374), and plan to present the results in a future paper that focuses on this point. We thank you for your valuable suggestion.

We have addressed each of the comments made by you, and we hope that our explanations and revisions are satisfactory.

Sincerely yours,

Manabu Okumura, Ph.D.

Department of Pharmacy

Faculty of Medicine

University of Miyazaki, Japan

Reviewer 2 Report

There are some comments.

There are limitations because of in vitro study. It is controversial whether HepG2 cells represent neoplastic cells. It would be better to describe limitations in the method and material (HepG2 cell) discussion.

It would be better to add light microscopic photos, if possible.

Please modify references according to the rules of IJMS.

It would be recommended to change title as follows:"Hepatocyte growth factor enhances antineoplastic effect of 5-fluorouracil by increasing UPP1 expression in HepG2 cells"

->"Hepatocyte growth factor enhances the antineoplastic effect of 5-fluorouracil by increasing expression of uridine phosphorylase 1 in HepG2 cells"

Author Response

August 1, 2022

Dear reviewer:

We are most grateful to you for the critical comments and useful suggestions on the original version of our manuscript entitled “Hepatocyte growth factor enhances antineoplastic effect of 5-fluorouracil by increasing UPP1 expression in HepG2 cells” (manuscript no.: ijms-1815911). These comments and suggestions have helped us to improve our manuscript considerably.

As indicated in the responses that follow, we have carefully considered the comments and suggestions made by you in the revised version of our manuscript.

Response to Reviewer 2 comments

Point 1: There are limitations because of in vitro study. It is controversial whether HepG2 cells represent neoplastic cells. It would be better to describe limitations in the method and material (HepG2 cell) discussion.

Response 1: As indicated by the reviewer, it is controversial whether HepG2 cells represent neoplastic cells. In this study, we evaluated the effects of HGF on the expression of critical metabolic enzymes involved in the antiproliferative effect of 5-FU. However, the direct proliferative effects of HGF on cancer cells could mask its secondary effects via the alteration of metabolism induced by HGF on the proliferation of these cells. In contrast, HepG2 cells hardly proliferate in response to HGF. Hence, we conducted experiments using HepG2 cells in this study. Therefore, we have retouched the materials and methods to explain on this point (Please see page 16, line 517). We thank you for your valuable suggestion.

Point 2: It would be better to add light microscopic photos, if possible.

Response 2: As suggested by the reviewer, we are also interested in the evaluation using light microscope. However, in this study, we have not taken light microscopic photos. We thank you for your valuable suggestion.

Point 3: Please modify references according to the rules of IJMS.

Response 3: In accordance with the reviewer’s suggestion, we have modified references according to the rules of IJMS (ACS Style Guide).

Point 4: It would be recommended to change title as follows:"Hepatocyte growth factor enhances antineoplastic effect of 5-fluorouracil by increasing UPP1 expression in HepG2 cells"

->"Hepatocyte growth factor enhances the antineoplastic effect of 5-fluorouracil by increasing expression of uridine phosphorylase 1 in HepG2 cells"

Response 4: In the "Instructions for Authors", when the title contains a protein name, it is instructed to use the abbreviated name. Should I use full name?  Please instruct us on how we should do. We will accordance with the reviewer’s instruction.

We have addressed each of the comments made by you, and we hope that our explanations and revisions are satisfactory.

Sincerely yours,

Manabu Okumura, Ph.D.

Department of Pharmacy

Faculty of Medicine

University of Miyazaki, Japan

Round 2

Reviewer 1 Report

The authors have answered partially to my comments. Although they do not provide experimental evidence, they have discussed the raised questions in the texts. However, they should rephrase/edit the newly inserted lines 374-377.

Author Response

August 5, 2022

Dear Reviewer,

Thank you very much for your highly insightful comment on the revised manuscript. We apologize that the first revised version of the manuscript was not satisfactory because we did not understand the exact meaning of your comment. We have revised the manuscript again based on your comment.

Response to Reviewer 1 comment

Point: The authors have answered partially to my comments. Although they do not provide experimental evidence, they have discussed the raised questions in the texts. However, they should rephrase/edit the newly inserted lines 374-377.

Response: In accordance with your suggestion, we have retouched the discussion to explain you indicated point (Please see page 13, line 374-384). However, we have not yet experimented with this point. We would like to evaluate it in our future research. Thank you for your valuable comment.

We hope that our explanations and revisions are satisfactory.

Sincerely yours,

Manabu Okumura, Ph.D.

Department of Pharmacy

Faculty of Medicine

University of Miyazaki, Japan